# Myofilament Alterations Associated with Human R14del-Phospholamban Cardiomyopathy

**DOI:** 10.3390/ijms24032675

**Published:** 2023-01-31

**Authors:** Mohit Kumar, Kobra Haghighi, Sheryl Koch, Jack Rubinstein, Francesca Stillitano, Roger J. Hajjar, Evangelia G. Kranias, Sakthivel Sadayappan

**Affiliations:** 1Department of Medicine, University of Cincinnati College of Medicine, Cincinnati, OH 45267, USA; 2Department of Pharmacology and Systems Physiology, University of Cincinnati College of Medicine, Cincinnati, OH 45267, USA; 3Division Heart and Lung, Department of Cardiology, University Medical Center Utrecht, 3584 CX Utrecht, The Netherlands; 4Phospholamban Heart Foundation, Postbus 66, 1775 ZH Middenmeer, The Netherlands

**Keywords:** R14del-phospholamban, myofilaments, contractility, omecamtiv mecarbil, males, females

## Abstract

Phospholamban (*PLN*) is a major regulator of cardiac contractility, and human mutations in this gene give rise to inherited cardiomyopathies. The deletion of Arginine 14 is the most-prevalent cardiomyopathy-related mutation, and it has been linked to arrhythmogenesis and early death. Studies in *PLN*-humanized mutant mice indicated an increased propensity to arrhythmias, but the underlying cellular mechanisms associated with R14del-*PLN* cardiac dysfunction in the absence of any apparent structural remodeling remain unclear. The present study addressed the specific role of myofilaments in the setting of R14del-*PLN* and the long-term effects of R14del-*PLN* in the heart. Maximal force was depressed in skinned cardiomyocytes from both left and right ventricles, but this effect was more pronounced in the right ventricle of R14del-*PLN* mice. In addition, the Ca^2+^ sensitivity of myofilaments was increased in both ventricles of mutant mice. However, the depressive effects of R14del-*PLN* on contractile parameters could be reversed with the positive inotropic drug omecamtiv mecarbil, a myosin activator. At 12 months of age, corresponding to the mean symptomatic age of R14del-*PLN* patients, contractile parameters and Ca^2+^ transients were significantly depressed in the right ventricular R14del-*PLN* cardiomyocytes. Echocardiography did not reveal any alterations in cardiac function or remodeling, although histological and electron microscopy analyses indicated subtle alterations in mutant hearts. These findings suggest that both aberrant myocyte calcium cycling and aberrant contractility remain specific to the right ventricle in the long term. In addition, altered myofilament activity is an early characteristic of R14del-*PLN* mutant hearts and the positive inotropic drug omecamtiv mecarbil may be beneficial in treating R14del-*PLN* cardiomyopathy.

## 1. Introduction

Arrhythmogenic cardiomyopathy (ACM, OMIM #609040) is an inherited disease characterized by frequent ventricular arrhythmias (VAs). In its early concealed form, it is a leading cause of sudden cardiac death (SCD) in young individuals, usually related to exercise and adrenergic stimulation. The main ACM-related genes encode desmosomal proteins such as desmoplakin (DSP), plakophilin-2 (*PKP2)*, desmoglein-2 (*DSG2)*, desmocollin-2 (*DSC2)* and plakoglobin (*JUP*), and non-desmosomal proteins such as desmin (*DES),* transmembrane protein 43 (*TMEM43)*, transforming growth factor β-3 (*TGFB3)*, lamin A/C (*LMNA)*, titin (*TTN)*, and αT-catenin (*CTNNA3*). However, in more recent years, mutations in calcium cycling genes (ryanodine receptor or *RyR2* and phospholamban or *PLN)* have been reported as causes of ACM [1,2,3] One of these mutations associated with deletion of NM_002667.5:c.40_42del (Mutalyzer 3), codon 14 (AGA), leading to deletion of Arginine (R) (R14del-*PLN*), was identified in human *PLN* [3,4]. *PLN* is a prominent regulator of SR Ca^2+^ cycling and contractility. In the dephosphorylated state, *PLN* is an inhibitor of Ca^2+^ affinity of sarcoendoplasmic reticulum Ca^2+^-ATPase (SERCA) in the heart, whereas phosphorylation during β-adrenergic stimulation relieves the inhibitory effects [1]. The R14del-*PLN* mutation is mainly characterized by low-voltage electrocardiograms (ECGs) at an early age and the development of dilated cardiomyopathy or arrhythmogenic right ventricular cardiomyopathy at a later stage [4,5]. Interestingly, the appearance of low-voltage ECGs is more common in females than males [6]. The onset of disease appears to be age-dependent, and symptoms develop most often in the fifth decade with a mean age at presentation ranging from 40 to 48 years [4,5,7].

The R14del-*PLN* founder mutation has been identified in a substantial percentage of Dutch patients, but today, this mutation has been found in several European countries, China, South Africa, the United States and Canada. The mutation is only found in the heterozygous state, and no homozygous patients have been identified. The haplotype containing this mutation was estimated to be 575 years old, showing that all R14del-*PLN* mutation carriers are distantly related [8,9]. Currently, no clinical studies or established treatments, other than standard heart-failure therapy or heart transplantation, have been available to patients with the R14del-*PLN* mutation, calling for the elucidation of the molecular and cellular pathways associated with arrhythmogenic cardiomyopathy.

Studies using cardiomyocytes derived from mutant R14del-*PLN* induced pluripotent stem cells (iPSC-CMs) indicated irregular Ca^2+^ waves and electrical instability [10]. An extension of these studies to engineered cardiac tissue revealed decreased contractility and increased susceptibility to triggered activity [11], consistent with super-inhibitory effects of this mutant on SERCA, a key protein in the calcium cycle of cardiomyocytes. However, a recent study reported that R14del-*PLN* may actually function as an enhancer of SR Ca^2+^ uptake in patient-specific pluripotent-stem-cell-derived cardiomyocytes [12]. Thus, it is suggested that further studies could result in a stronger understanding of the underlying mechanisms of R14del-*PLN*, especially in vivo. Following up on this notion, we generated humanized *PLN* mice since there is a difference at position 27 between mouse and human *PLN*. All species contain asparagine (Asn), while human *PLN* contains lysine (Lys) at amino acid 27. This substitution results in a different N-terminal helix of the human *PLN*, as revealed by NMR studies, and increased inhibition of SERCA with a resultant depressed cardiac function [13]. Furthermore, Asn27 in *PLN* was reported to interact with Leu321 in *SERCA*, and alterations in this residue are expected to have significant effects on the Ca^2+^ affinity of SERCA. Indeed, R14del-*PLN* exhibits an increased association with both SERCA and Hax-1, the binding partner of *PLN* that increases its inhibitory activity. These findings suggest a super-inhibitory effect of mutant *PLN* on SR Ca^2+^ transport activity [14]. Parallel studies in mouse models with a knock-in of human wild-type *PLN* (WT-*PLN*) or a heterozygous *PLN* (R14del-*PLN*) mutation indicated that R14del-*PLN* was associated with an increased propensity to arrhythmias under stress conditions in vivo. The cellular mechanisms included aberrant sarcoplasmic reticulum Ca^2+^ cycling, increases in diastolic Ca^2+^, and depressed contractile parameters in right ventricular myocytes (RV) in association with increased Ca^2+^ leak and triggered beats [15]. Importantly, disruption of the human R14del-*PLN* allele with AAV9-CRISPR/Cas9 improved cardiac function and reduced ventricular tachycardia susceptibility in the humanized R14del-*PLN* mice [16]. However, the detrimental effects of R14del-*PLN* might be mediated by pathways other than impaired SR Ca^2+^ cycling.

In the current study, we investigated the role of myofilaments in R14del-*PLN* pathology. The contractile properties of cardiomyocytes are determined by the regulation of myofilament proteins and their sensitivity to calcium. Studies have shown that changes in Ca^2+^ handling and myofilament proteins resulted in contractile dysfunction as an early sign of developing heart failure and cardiac arrhythmias [17,18,19]. Thus, the present work aimed to determine the myofilament properties of cardiomyocytes in both right (RV) and left ventricular (LV) cardiomyocytes from R14del-*PLN* mice. Studies in skinned fibers of young mice indicated decreases in the maximal velocity that were more pronounced in RV myocytes. In addition, myofilament Ca^2+^ sensitization was increased, and these alterations were similar between LV and RV skinned myocytes. To determine the long-term effects of R14del-*PLN* mutation, we extended characterization studies to mice at 12 months of age, corresponding to the critical age of human patients [4,5,7]. Overall, depressed contractility and calcium transients remained specific to RV cardiomyocytes, while there were histological and ultrastructural abnormalities in mutant hearts.

## 2. Results

### 2.1. Decreased Maximal Force in R14del-PLN Cardiomyocytes

A previous study indicated that R14del-*PLN* mutation resulted in depressed Ca^2+^ kinetics and contractile parameters specific to RV myocytes at 3 months of age [15]. To determine whether R14del-*PLN* was associated with any alterations in myofilament function, we used skinned myocytes from right and left ventricles of 3-month-old mice and measured the level of maximal force generation (F_max_), active force–[Ca] relationships (pCa–force), myofilament calcium sensitivity (EC_50_) and Hill coefficient (nH), an index of activation of cooperativity at varying calcium concentrations (pCa 10.0 to pCa 4.5) (Figure 1). Strikingly, maximal force from both right and left ventricles was significantly decreased in mutant myocytes compared to WT-*PLN* controls, but this decrease was more pronounced in the right ventricle (Figure 1A,B,E,F). Further, myofilament calcium sensitivity in both right and left ventricles of mutant skinned myocytes was significantly increased compared to WT-*PLN* controls (Figure 1C,D,G,H). However, the Hill coefficient (nH: ~2) was not altered. These data suggest that R14del-*PLN* myocytes need less calcium to reach 50% of the maximal force compared to WT-*PLN* myocytes, which may be an important compensatory mechanism.

### 2.2. Omecamtiv Mecarbil as a Potential Drug to Treat R14del-PLN-Mediated Contractile Dysfunction

Based on the observed effects of R14del-*PLN* on contractile parameters, we examined whether the myosin-specific pharmaceutical activator omecamtiv mecarbil (OM, Figure 2) could reverse R14del-*PLN*-mediated contractile dysfunction and improve the contractile properties of 3-month-old cardiomyocytes in vitro. We also used a myosin-specific pharmaceutical inhibitor, Mavacamten (MYK-461, Figure 3), to determine any changes in contractile force. Contractile parameters, including the amplitude and duration of sarcomere shortening, contraction and relaxation velocity, and resting sarcomere length, were measured. Parameters of Ca^2+^ transients, including the amplitude at systole and duration of Ca^2+^ decay (tau), were also simultaneously measured. As expected, baseline contractile parameters from right ventricles of R14del-*PLN* mice were reduced, but no alterations in LV myocytes were observed. Treatment of cardiomyocytes with OM significantly improved the fractional shortening (Figure 2A), contraction velocity (+dL/dt; Figure 2B), and relaxation velocity (−dL/dt; Figure 2C) in RV mutant myocytes, but it had no effects on LV myocytes. However, no parallel stimulatory effects on Ca^2+^ transients of RV mutant myocytes were noted (Figure 2D,E) compared to controls. As a control experiment, we tested the impact of a myosin-specific pharmaceutical inhibitor, mavacamten (MYK-461), on the contractile properties and Ca^2+^ transients in WT-*PLN* and R14del-*PLN* myocytes. As expected, MYK-461 decreased the contractility in both WT-*PLN* and R14del-*PLN* myocytes similarly, but without affecting Ca^2+^ transients (Figure 3A–E), indicating that inhibition of myosin in the cardiomyocytes of R14del-*PLN* mice has no benefits. These findings are consistent with our previous data indicating that OM directly enhances myocyte contractility without affecting Ca^2+^ homeostasis [20].

### 2.3. Decreased Contractility and Calcium Transients in R14del-PLN Cardiomyocytes at 12 Months

In R14del-*PLN* patients, symptoms develop most often in the fifth decade with a mean age at presentation ranging from 40 to 48 years [4,5,7]. In addition, the onset of disease appears to indicate a slightly higher frequency in males [7]. Thus, we examined cardiomyocyte contractile parameters and Ca^2+^ kinetics, as well as sex-dependent effects, in the humanized models at 12 months of age, corresponding to the age of patients who typically present with symptoms. In male mice, LV contractile parameters were similar between WT-*PLN* and mutant hearts (Figure 4A–C). However, fractional shortening (FS) and rates of contraction (+dL/dt) and relaxation (−dL/dt) were all significantly reduced in RV mutant myocytes compared to WT-*PLN* (Figure 4A–C). These decreases in mutant RV contractile parameters were similar between males and females (Figure 4D–F). Interestingly, mutant females also exhibited depressed rates of contraction and relaxation in LV myocytes (Figure 4D–F). Furthermore, intracellular Ca^2+^-cycling parameters were assessed in isolated cardiomyocytes using the Fura-2 AM fluorescence indicator (2 µM). R14del-*PLN* elicited decreases in the Ca^2+^ peak of transients and prolongation of the time constant for Ca^2+^ decay (tau) in male RV myocytes (Figure 5A–C). Examination in females revealed similarly depressed parameters specifically in RV myocytes (Figure 5D,E). However, no parallel alterations in LV mutant cells from either males or females were observed (Figure 5).

### 2.4. Normal Cardiac Function and Geometry in R14del-PLN Mice at 12 Months of Age

Echocardiography assessment indicated similar ejection fraction, fractional shortening and cardiac output between these groups (Figure 6B–D), suggesting that in vivo neurohormonal factors may mask the contractile differences in RV myocytes between WTs and mutants. No differences in geometrical parameters or remodeling occurred at 12 months of age, as evidenced by the left ventricular end-systolic diameter (LVESD) and left ventricular end-diastolic diameter (LVEDD) between WT-*PLN* and R14del-*PLN* hearts (Figure 6E,F). Furthermore, these parameters were similar between males and females (Figure 6). Therefore, R14del-*PLN* did not affect in vivo cardiac function or remodeling up to 12 months.

### 2.5. Increased Fibrosis and Altered Sarcomere Structures in R14del-PLN Hearts at 12 Months of Age

To assess cardiac pathology in 12-month-old WT-*PLN* and R14del-*PLN* hearts, we used Picrosirius Red staining of fixed cardiac tissue sections. Pathological analysis revealed significant interstitial fibrosis in both LV and RV tissues from R14del-*PLN* male and female mice (Figure 7A,C). Quantification of collagen levels indicated twofold increases in mutant males and females compared to WTs (Figure 7B,D). In addition, ultrastructural analysis of longitudinal sections from LV (top) and RV (bottom), using transmission electron microscopy, revealed abnormalities in both males and females from R14del-*PLN* hearts. Irregular, disarrayed and widened Z-disc patterns (white box) were observed, as well as streaming and smearing of the Z-disc material, in R14del-*PLN* ventricles (white boxes) (Figure 8A,B). Furthermore, a higher number of lipid droplets were observed in mutant ventricles from both male and female hearts (Figure 9A,B), similar to previous findings in human R14del-*PLN* patients [21]. These results suggest that structural abnormalities may contribute to the ventricular pro-arrhythmic phenotype in R14del-*PLN* hearts under stress conditions.

## 3. Discussion

The present study aimed to demonstrate that human R14del-*PLN* is associated with myofilament dysfunction, potentially one of the underlying cellular mechanisms contributing to the pathogenesis of cardiomyopathy. Thus, besides the primary role of mutant *PLN* to inhibit SERCA activity and SR Ca^2+^ cycling [4,14], it also affects the contractile protein function, resulting in further depression of cardiac contractility. In this study, we used humanized-*PLN* mouse models with only one amino acid difference between human and all other species, possibly affecting the regulatory effects of *PLN* on SERCA. It has been suggested that the inhibitory effects of *PLN* involve amino acids 21 to 30 containing a single positive charge or Arg25 in most species. However, Lys27 replaces Asn27 in humans, resulting in two positive charges, which may alter the secondary and tertiary structure of *PLN*. Indeed, NMR analysis of AA 1–36 indicated that the spatial conformation of human *PLN* is significantly different from the *PLN* of mouse or any other mammalian species due to the association with looser packing of the human peptide, which may influence the interaction between *PLN* and SERCA.

Myofilament calcium sensitivity was increased in mutant cardiomyocytes, and the increase was more pronounced in RV cells. This may serve as an early compensatory alteration in R14del-*PLN* hearts to improve contractility in the face of decreased SERCA activity and SR Ca^2+^ load, especially in RV myocytes [15]. However, although such increases in calcium affinity would be beneficial in the short term, they are expected to damage the force-generating capacity of myofilaments over time. In the long term, these changes could lead to chronic increases in systolic and diastolic Ca^2+^ and the activation of CaMKII, which phosphorylates the ryanodine receptors. This, in turn, promotes SR Ca^2+^ leak and cardiac arrhythmias, as evidenced in patients carrying the R14del-*PLN* mutation [4,5]. In addition, previous studies reported that myofilament calcium sensitization increases the susceptibility to ventricular arrhythmias [18,19]. Indeed, there was a direct correlation between the degree of calcium sensitization and the risk for ventricular arrhythmias in *TNNT2* mutant mice [18,19]. We also observed a decreased myofilament force in R14del-*PLN* myocytes, which may be the result of calcium sensitization and may reflect altered cross-bridge kinetics or decreases in the fractional volume occupied by contractile machinery. Like our findings, a recent study in human failing hearts indicated a decreased maximal force and increased calcium sensitivity of contraction in the ventricles [22].

The depressive effects of R14del-*PLN* on the contractile parameters of RV myocytes were prevented by OM, while MYK-461 had no effects. OM is a cardiac myosin activator and increases the coupling between myocardial actin and myosin without increasing myocardial oxygen consumption [23]. On the other hand, MYK-461, which is a selective inhibitor of cardiac myosin, had no effect on R14del-*PLN*. OM differs from other inotropes that augment myocardial force by elevating intracellular calcium levels. Initial clinical studies with OM showed positive inotropic effects and reduced left ventricular systolic and diastolic volumes [23,24,25,26]. However, OM induces continuous activation of resting myosin ATPase and increases myocardial oxygen consumption, which may limit its clinical benefits in current treatment options.

In human patients, the clinical characteristics of R14del-*PLN* pathology are most evident by mid-age, albeit with incomplete penetrance, and some R14del-*PLN* carriers remain asymptomatic through life, while others require cardiac transplantation by mid-age [4,5,7]. The mid-age of humans corresponds to about 12 months of age in mice. Characterization of humanized mice at 12 months indicated impaired RV-specific myocyte contractile parameters and calcium kinetics. Interestingly, the rates of contraction and relaxation were also depressed in LV myocytes from female mice, while there were no parallel decreases in Ca^2+^ kinetics. This may indicate additional compensation by the myofilament activity. Nevertheless, the assessment of cardiac function at the whole-animal level did not reflect any differences between males and females. These findings in the humanized mouse model do not reflect previous observations in human R14del-*PLN* patients, indicating an earlier onset in males [6,7]. In addition, studies in failing human hearts showed a gender-specific depression in SR Ca^2+^-cycling protein activity [27]. Further characterization using echocardiography did not reveal any alterations in cardiac function or geometrical parameters at 12 months of age, indicating that altered Ca^2+^ cycling and intrinsic contractility of RV cardiomyocytes were not sufficient to elicit changes at the intact-animal level. In addition, there were no spontaneous premature ventricular contractions (PVCs) under basal conditions, while catecholaminergic stress induced a high burden of PVCs and ventricular tachycardia, similar to younger mice [3]. However, the depressed calcium homeostasis at the cardiomyocyte level probably contributed to cellular stress inducing a pathological cardiac myocyte-fibroblast crosstalk. This may occur through both direct contact and paracrine mechanisms leading to fibroblast activation and ventricular fibrosis [28].

Indeed, extensive fibrosis and lipid droplets were observed in mouse mutant hearts, similar to findings in patients who showed cardiac fibrosis and fibrofatty replacement, even at the presymptomatic stage [29,30,31]. This may serve as a substrate for cardiac arrhythmias. In addition, R14del-*PLN* mouse hearts exhibited alterations in Z-disc formation or organization, consistent with previous proteomic analyses that indicated alterations in myofilament proteins of these mouse hearts [16]. Accumulation of lipid droplets and disturbed mitochondrial integrity, as observed in R14del-*PLN* patients’ hearts [21], were also reported. Indeed, increases in SR Ca^2+^ leak have been previously shown to trigger mitochondrial dysfunction [31]. These ultrastructural alterations may provide valuable insights towards targeted therapy to prevent or reverse pathological features associated with R14del-*PLN* pathology.

In summary, our findings in humanized R14del-*PLN* mice indicate decreases in the maximal force with parallel increases in the Ca^2+^ affinity of myofilaments which may contribute to overall impaired Ca^2+^ cycling and increased propensity to arrhythmias in mutant mice. Interestingly, the aberrant Ca^2+^ cycling and contractility specific to right ventricular myocytes at a young age persisted up to twelve months of age, which is an age that corresponds to cardiac pathology in R14del-*PLN* patients. The cellular alterations were similar between male and female mice and were associated with interstitial fibrosis and altered cellular organization that was more pronounced in mutant RV. These findings highlight the importance of understanding cellular defects associated with R14del-*PLN*, providing the opportunity for targeted therapy.

## 4. Materials and Methods

### 4.1. Humanized WT-PLN and R14del-PLN Knock-in Mice

Mice harboring human WT-*PLN* and the R14del-*PLN* coding sequence were created by inserting the LoxP-H2B-GFP- 4XpolyA-FRT-Neo-FRT-LoxP-h*PLN*^WT/R14del^ cassette into the *PLN* start codon at exon 2, as previously described [15,32]. The handling and maintenance of animals were approved by the Ethics Committee of the University of Cincinnati. The investigation followed the “Guide for the Care and Use of Laboratory Animals” of the National Institutes of Health.

### 4.2. Measurements of Cardiomyocyte Contraction Mechanics and Ca^2+^ Kinetics

Cell contractility and Ca^2+^ transients were measured simultaneously at room temperature (22–23 °C), as previously described [20,33,34,35,36]. Briefly, under basal conditions, cardiomyocytes were incubated with 1.0 μM Ca^2+^-sensitive Fura-2 dye (Invitrogen, Waltham, MA, USA) for 15 min at room temperature and then washed with fresh regular Tyrode’s solution for at least 10 min. Then, cardiomyocytes were placed in a perfusion chamber adapted to the stage of an inverted Nikon eclipse TE2000-U fluorescence microscope. Cells were superfused (Warner Instruments, Boston, MA, USA, six-channel valve controller) with Tyrode’s buffer at room temperature. Sarcomere shortening was assessed using a video-based sarcomere length detection system (Ion-Optix, Milton, MA, USA). To measure intracellular Ca^2+^ transients, Fura-2 fluorescence was excited at 340 and 380 nm and acquired at an emission wavelength of 515 ± 10 nm using a spectrofluorometer (IonOptix, Milton, MA, USA). The emission field was restricted to a single cell with the aid of an adjustable window. Simultaneous measurements of mechanics and Ca^2+^ kinetics were performed at 0.5 Hz in the absence and presence of 100 nmol/L omecamtiv mecarbil (OM) [23], which augments cardiac contractility, and 250 nM mavacamten, a cardiac myosin inhibitor (MYK-461) [24]. Both male and female mice were utilized. Data were analyzed with IonOptix LLC analyzing software (Version 7).

### 4.3. Measurements of pCa–Force and Myofilament Ca^2+^ Muscle Sensitivity Using Skinned Myocytes

Permeabilized cardiomyocytes from male WT-*PLN* or R14del-*PLN* mice were prepared in relaxing solution from frozen tissue and analyzed for pCa–force, as previously described [25]. In brief, 20–30 mg of frozen heart tissue were homogenized twice at 5000 rpm for 7 s and filtered through a 70-µm cell strainer, followed by centrifugation at 100× *g* for 3 min at 4 °C. To remove the cell membrane, a skinning process was performed by resuspending pelleted cells in relaxing solution [97.92 mM KOH, 6.24 mM ATP, 10 mM EGTA, 10 mM Na2CrP, 47.58 mM potassium propionate, 100 mM BES, 6.54 mM MgCl_2_, 1 mM DTT and 100 μM protease cocktail (Sigma-Aldrich, St. Louis, MO, USA)] with 1% Triton-X. The cells were kept on a rotating shaker at room temperature for 10–15 min. Thereafter, the cells were washed twice with two centrifugations at 100× *g* for 3 min at 4 °C with the relaxing solution to remove Triton. Using the 1600A Permeabilized Myocyte System (Aurora Scientific Inc., Aurora, ON, Canada), skinned myocytes with regular striation patterns were selected and attached to a force transducer (Aurora Scientific Inc., ON, Canada) and a high-speed piezo translator (Thorlabs, Newton, NJ, USA) using optical glue (Norland, Cranbury, NJ, USA). Cells were then exposed to maximum saturating [Ca^2+^] activating solution (pCa 4.5) at the beginning and end of the experimental protocol to test the strength and rundown of cell attachment. Isometric force development was recorded at a sarcomere length of 1.9 µm using the activating solution with varying calcium concentrations (pCa 10.0 to pCa 4.5), while sarcomere length was continuously monitored through Aurora’s High-Speed Video Sarcomere Length (HVSL) measurement system. The recorded total isometric force, after correcting the rundown, was normalized to the cross-sectional area that was calculated by buckling the attached myocyte and measuring the dimensions of the elliptical cell’s two sides. Data were acquired using Aurora’s 600A real data acquisition and analysis system. Force–pCa curves were fit using the modified Hill equation (Force/Force_max_ = [Ca^2+^]^n^/(pCa_50_^n^ + [Ca^2+^]^n^), where n is the Hill-slope.

### 4.4. Assessment of Cardiac Function with Echocardiography

Echocardiographic analysis was performed to assess cardiac function using the Vevo 5 2100 Ultrasound system (Visualsonics, Toronto, ON, Canada) as previously described [4]. Briefly, following anesthetization with isoflurane (1.5–2%), M-mode and B-mode images were obtained from a parasternal long-axis view between 2 and 10 mm in depth. Images were analyzed using Vevostrain software (Vevo 2100, v1.1.1 B1455, Visualsonics, Toronto, ON, Canada). All measurements were performed in a blinded manner that included left ventricular (LV) septum, posterior wall and chamber size at end-diastolic (LVEDD) and end-systolic diameters (LVESD). From these values, the FS was obtained with the software following the formula FS = (LVEDD − LVESD/LVEDD) × 100 and EF was calculated via the Teicholz formula by converting the chamber diameter [37,38].

### 4.5. Histopathological Analysis

Hearts were excised from anesthetized mice (Euthasol, 200 mg/kg i.p., Virbac AH, Inc., Fort Worth, TX, USA), immediately fixed in 10% neutral buffered formalin (Sigma, Saint Louis, MO, USA) and processed as previously described [15]. Briefly, for histology studies, paraffin-embedded hearts were sectioned (4 μm), deparaffinized, rehydrated, incubated in 0.1% Sirius red in saturated picric acid for 1 h and then washed in acidified water. Paraffin-embedment and slide preparation were carried out by the University of Cincinnati Pathology Department Core Facility. Images were acquired on a Leica DMi8 Widefield microscope with a 20× objective lens using LASX software (Leica Microsystems, 9435 Heerbrugg, Switzerland). The analysis of fibrosis levels in cross sections of LV and RV was performed by calculating the collagen fiber staining with Image J (version 1.48v; National Institutes of Health, Bethesda, Maryland, MD, USA).

### 4.6. Ultrastructure Analysis of Cardiomyocytes with Electron Microscopy

For transmission electron microscopy (TEM), RV and LV tissue samples were cut (1 mm^2^), fixed in 2.5% glutaraldehyde in 0.15 M sodium cacodylate buffer, post-fixed in 1% osmium tetroxide and processed through a series of ethanol alcohol dehydrations at concentrations of 25%, 50%, 75%, 95% and twice at 100%, incubating at 10-min increments for each step. Then samples were infiltrated and embedded in LX-112 resin. After polymerization at 60 degrees for three days, ultrathin sections (120 nm) were cut using a Leica EM UC7 ultramicrotome and counterstained in 2% aqueous uranyl acetate and Reynold’s lead citrate. Sample preparation and imaging were performed by the Electron Microscopy CORE facility at Cincinnati Children’s Hospital. Images were taken with a transmission electron microscope (Hitachi H-7650, Tokyo, Japan) equipped with a digital camera (Biosprint 16).

### 4.7. Statistical Analysis

Data are expressed as mean ± standard deviation of the mean (SD) for the number of mice, hearts or myocytes, and statistical analyses were performed with two-tailed unpaired *t*-tests using GraphPad Prism, version 8.4.3. All myocyte data were generated using LV and RV cells from the same mouse on any specific day. Grubbs and ROUT tests were performed to identify statistically significant outliers, and outliers were removed if appropriate. A *p*-value of <0.05 was considered statistically significant.

## Figures and Tables

**Figure 1 ijms-24-02675-f001:**
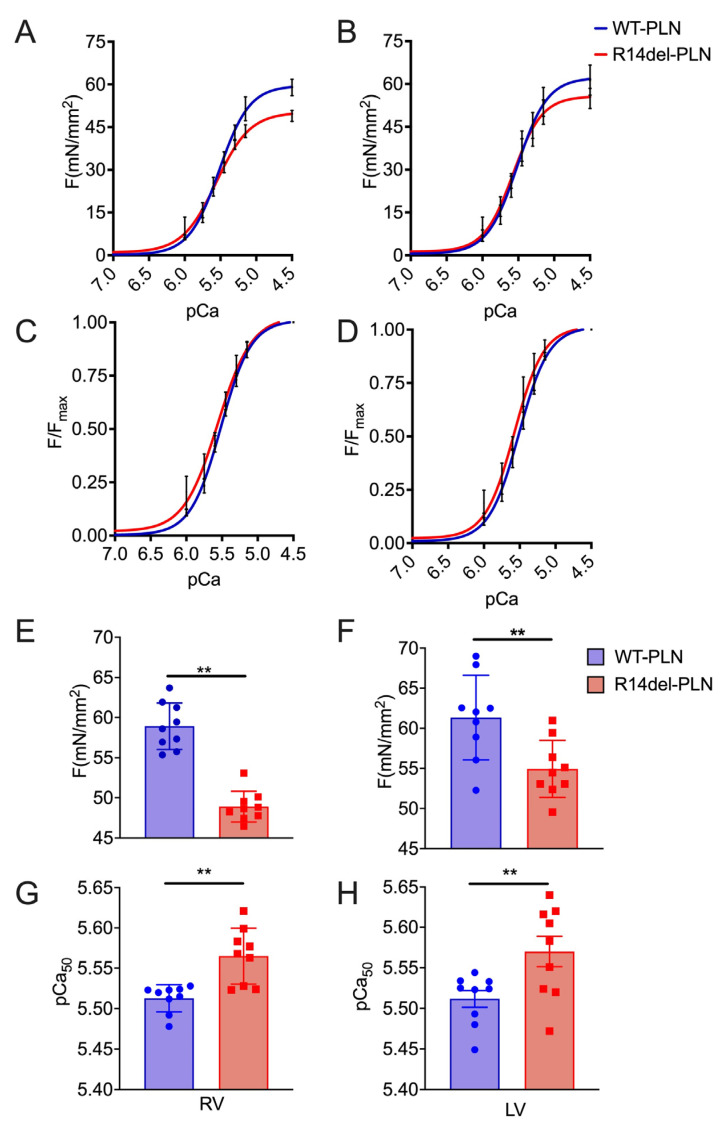
Force–pCa relationship in skinned cardiomyocytes. (**A**,**E**) Skinned RV myocytes from R14del-*PLN* hearts (48.9 ± 0.63 mN/mm^2^, n = 9 cells/3 hearts) showed a significant decrease in maximal force development compared with WT-*PLN* RV myocytes (58.92 ± 0.96 mN/mm^2^, n = 9 cells/3 hearts). (**B**,**F**) Similarly, LV skinned myocytes from R14del (54.94 ± 1.18 mN/mm^2^, n = 9 cells/3 hearts) showed a significant decrease in maximal force development, compared with WT-*PLN* LV skinned myocytes (61.34 ± 1.75 mN/mm^2^, n = 9 cells/3 hearts). Calcium sensitivity of force development, expressed as pCa_50_, was significantly increased in both RV (**C**,**G**) and LV (**D**,**H**) R14del myocytes compared to WT-*PLN* cardiomyocytes. mN = millinewtons. Data are expressed as mean ± SD, and statistical analyses were performed with Student’s unpaired *t*-test. ** *p* ≤ 0.01 vs. WT-*PLN*.

**Figure 2 ijms-24-02675-f002:**
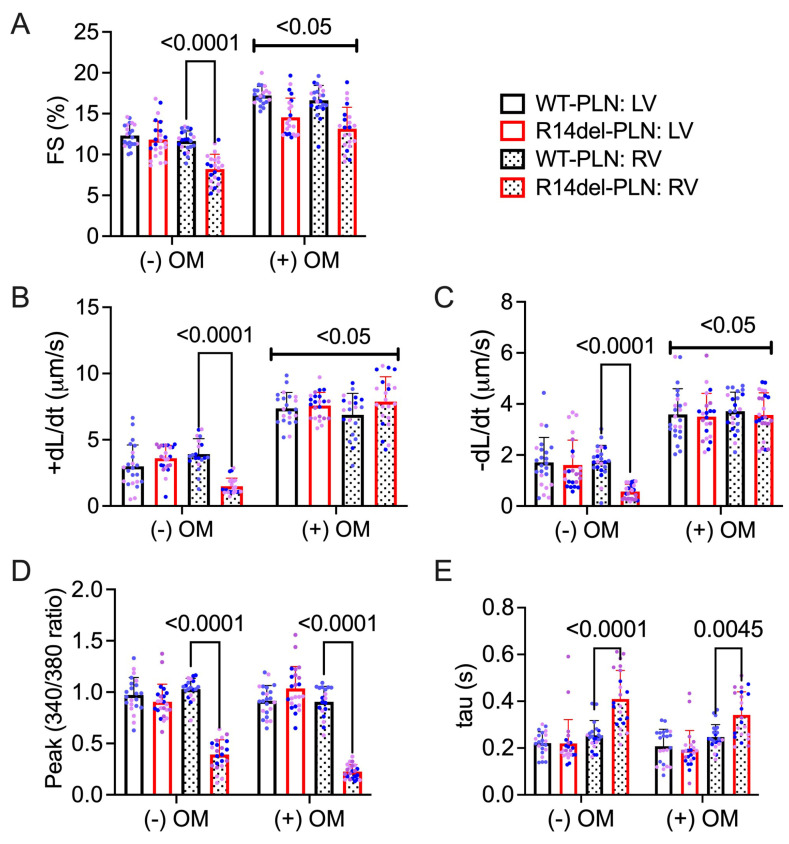
A positive cardiac inotropic drug, omecamtiv mecarbil (OM), improves contractility mechanics in R14del-*PLN* right ventricular skinned myocytes without affecting calcium handling. (**A**) Fractional shortening (FS) in the absence and presence of 100 nM myosin activator OM at 0.5 Hz. (**B**) Contraction velocity of sarcomere in the absence and presence of OM. (**C**) Relaxation velocity of sarcomere in the absence and presence of OM. (**D**) Ca^2+^ transient amplitude is calculated with the Fura-2 ratio (340:380 nm) in the absence and presence of 100 nM OM at 0.5 Hz with 1.8 mM Ca. (**E**) Relaxation time constant (tau) of calcium transient in the absence and presence of OM is presented. N = 3 hearts (20–22 cells)/group. Blue and pink dots represent male and female data points, respectively. Data are expressed as mean ± SD, and statistical analyses were performed in all groups using ordinary two-way ANOVA followed by Tukey’s multiple comparison test.

**Figure 3 ijms-24-02675-f003:**
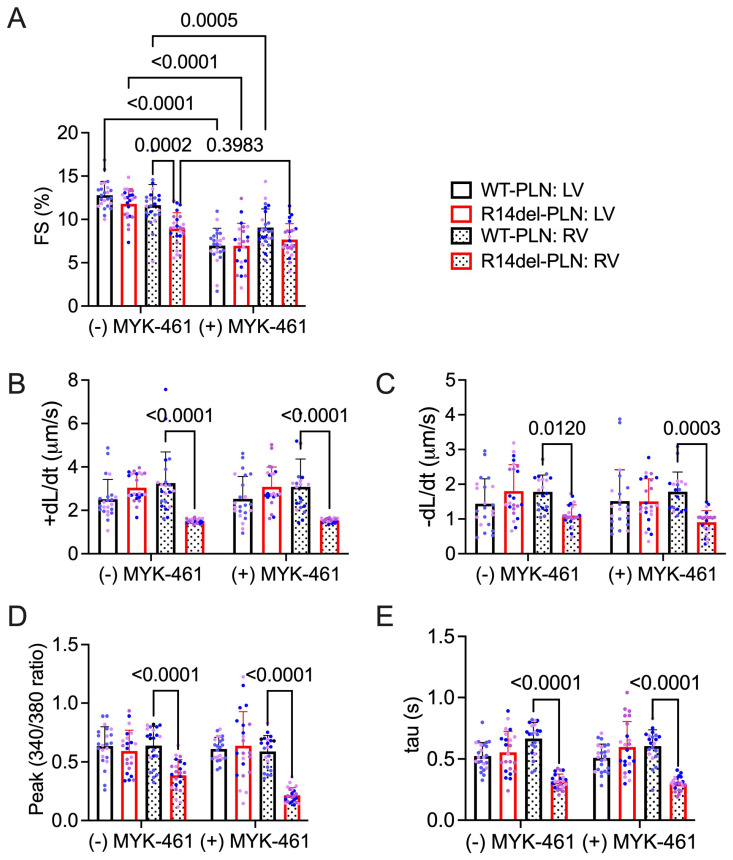
The myosin inhibitor MYK-461 depresses contractility with no effect on calcium handling. (**A**) Fractional shortening (FS) in the absence and presence of 250 nM MYK-461 at 0.5 Hz. (**B**) Contraction velocity of sarcomere in the absence and presence of MYK-461. (**C**) Relaxation velocity of sarcomere in the absence and presence of MYK-461. (**D**) Ca^2+^ transient amplitude as indicated by the Fura-2 ratio (340:380 nm) in the absence and presence of 250 nM MYK-461 at 0.5 Hz with 1.8 mM Ca^2+^ and (**E**) the relaxation time constant (tau) of calcium transient in the absence and presence of MYK-461. N = 3 hearts (20–22 cells)/group. Blue and pink dots represent male and female data points, respectively. Data are expressed as mean ± SD, and statistical analyses were performed in all groups using ordinary two-way ANOVA followed by Tukey’s multiple comparison test.

**Figure 4 ijms-24-02675-f004:**
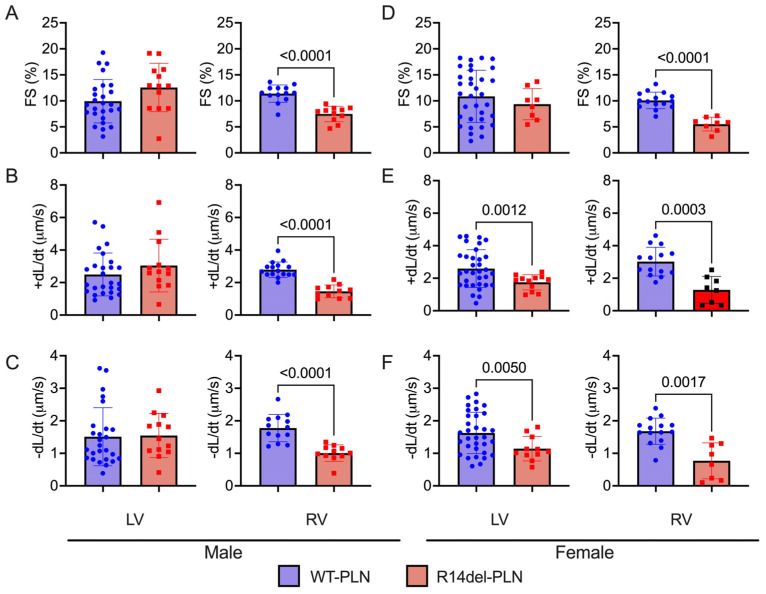
Contractile parameters in isolated cardiomyocytes from 12-month-old male (left panels) and female (right panels) WT-*PLN* and R14del-*PLN* mice. (**A**,**D**) Fractional shortening (FS) in % at 0.5 Hz. (**B**,**E**) Maximum rates of contraction velocity (+dL/dt). (**C**,**F**) Maximum rates of relaxation velocity (−dL/dt). n = 23 LV and 15 RV cells for WT males (N = 3 hearts); n = 37 LV and 12 RV cells for WT females (N = 5 hearts). n = 13 LV and 11 RV cells for R14del-*PLN* males (N = 3 hearts); n = 8 LV and 8 RV cells for R14del-*PLN* females (N = 3 hearts). Data are expressed as mean ± SD for the number of cells, and statistical analyses were performed with Student’s unpaired *t*-test.

**Figure 5 ijms-24-02675-f005:**
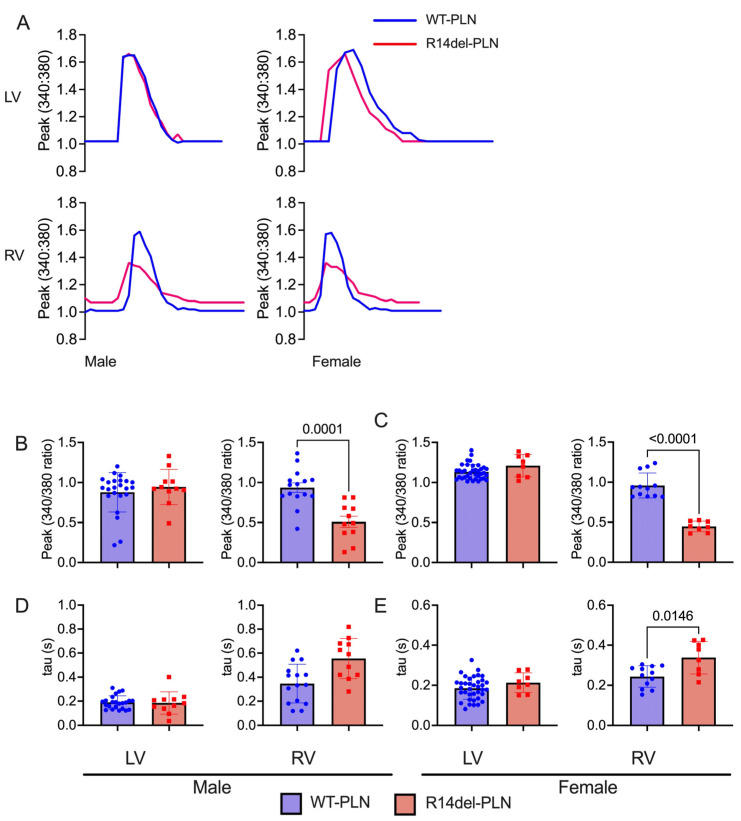
Ca^2+^ kinetics in isolated cardiomyocytes from 12-month-old male and female WT-*PLN* and R14del-*PLN* mice. (**A**) Shown are representative Ca^2+^ transient tracings of WT-*PLN* and R14del-*PLN*. (**B**,**C**) Ca^2+^ transient amplitude, as indicated by the Fura-2 ratio (340:380 nm) at 0.5 Hz with 1.8 mM Ca. (**D**,**E**) Relaxation time constant (tau) of calcium transient decay. n = 23 LV and 15 RV cells for WT-*PLN* males (N = 3 hearts); n = 37 LV and 12 RV cells for WT-*PLN* females (N = 5 hearts). n = 13 LV and 11 RV cells for R14del-*PLN* males (N = 3 hearts); n = 8 LV and 8 RV cells for R14del-*PLN* females (N = 3 hearts). Data are expressed as mean ± SD for the number of cells, and statistical analyses were performed with Student’s unpaired *t*-test.

**Figure 6 ijms-24-02675-f006:**
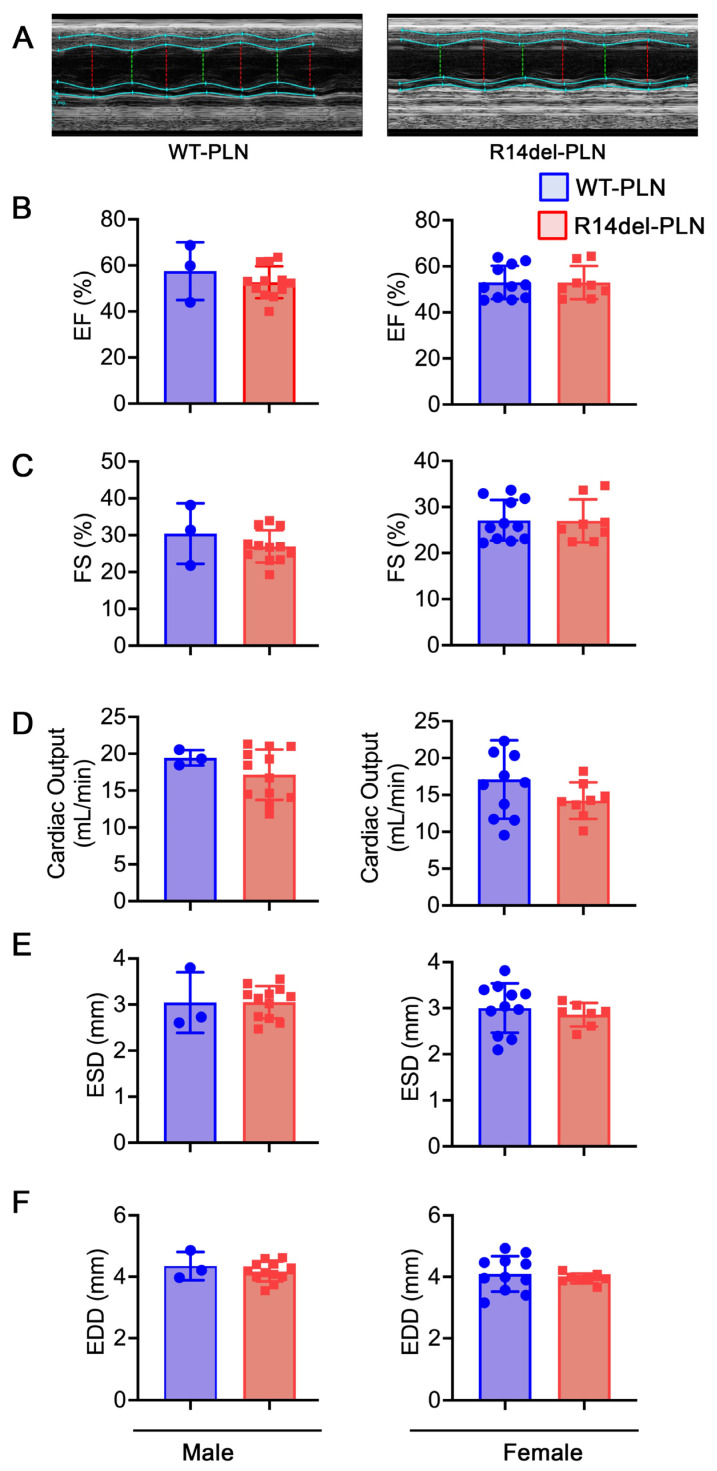
R14del-*PLN* mice exhibit no evidence of cardiac dysfunction or remodeling. (**A**) Sample M-Mode echocardiograms of male WT-*PLN* and R14del-*PLN* mice at 12 months of age. Quantitative analysis of echocardiographic parameters included (**B**) Ejection fraction (EF%); (**C**) Fractional shortening (FS%); (**D**) Cardiac Output (CO); (**E**) Left ventricular internal dimension, diastole (LVID;d); and (**F**) Left ventricular internal dimension, systole (LVID;s). Values represent mean ± SD; n = 3 for WT-*PLN* males; n = 12 for WT-*PLN* females; n = 12 for R14del-*PLN* males; and n = 8 for R14del-*PLN* females.

**Figure 7 ijms-24-02675-f007:**
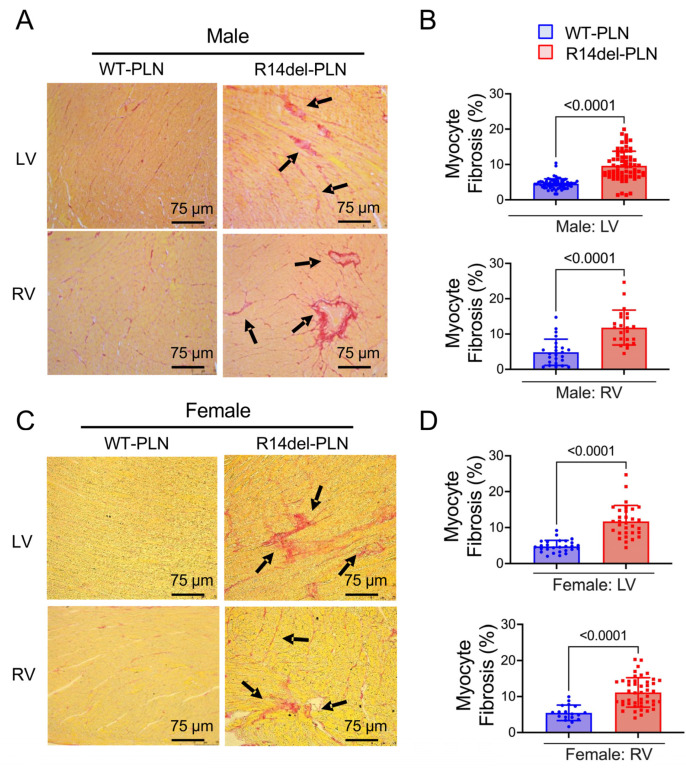
Histological evaluations of left ventricle (LV) and right ventricle (RV) from 12-month-old male and female R14del-*PLN* and WT-*PLN* mice. (**A**,**C**) Representative images of ventricular sections stained with Picrosirius red to evaluate the level of fibrosis by observing collagen deposition in LV and RV tissues from WT-*PLN* and R14del-*PLN* in paraffin-embedded sections (4 μm). Picrosirius red staining of LV and RV from R14del-*PLN* sections showed significant interstitial (black arrows) and perivascular (arrows) deposition of collagen fibers (red) compared with WT-*PLN* (left panels). Scale bar = 75 μm, 20× magnification views. (**B**,**D**) Quantitative analysis of ventricular fibrosis (expressed as a percentage of total area) in WT-*PLN* and R14del-*PLN* hearts (right panel). Data are expressed as mean ± SD for the myocyte fibrosis. n = 5 per group.

**Figure 8 ijms-24-02675-f008:**
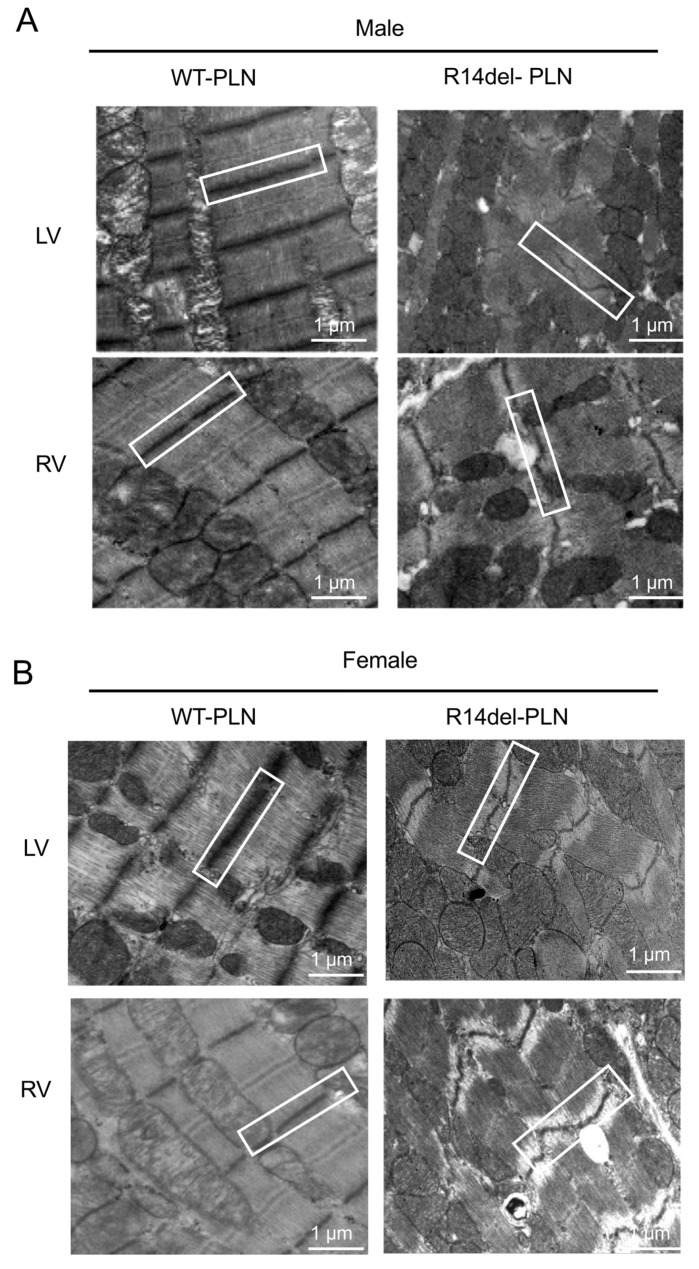
Transmission electron microscopy (TEM) evaluations of left ventricle (LV top panels) and right ventricle (RV bottom panels) from 12-month-old male (**A**) and female (**B**) WT-*PLN* and R14del-*PLN* mice. Representative images showed irregular/disarrayed, widened Z-disc patterns as well as streaming and smearing of the Z-disc material in R14del ventricles (white box). N = 4 per group. Scale bars: 1 μm, magnification 6000×.

**Figure 9 ijms-24-02675-f009:**
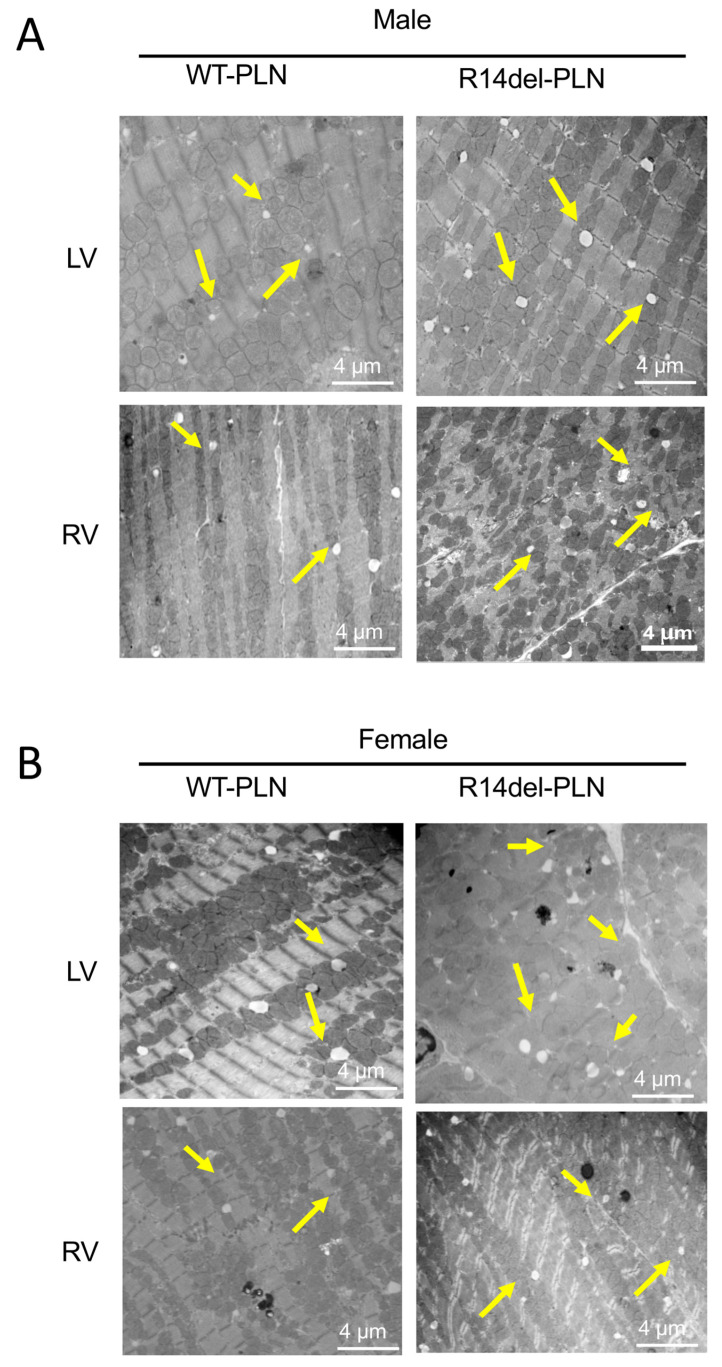
Transmission electron microscopy (TEM) evaluations of left ventricle (LV top panels) and right ventricle (RV bottom panels) from 12-month-old male (**A**) and female (**B**) WT-*PLN* and R14del-*PLN* mice. Ultrastructural images from longitudinal sections of LV and RV showed lipid droplets (yellow arrows), with a higher number in R14del-*PLN* ventricles. N = 4 per group. Scale bar: 4 μm, magnification 15,000×.

## Data Availability

The data presented in this study are available on request from the corresponding authors, Drs. Kranias and Sadayappan. The data are not publicly available due to restrictions, e.g., privacy, intellectual properties, or ethics.

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
