# Peer review of "Myofilament Alterations Associated with Human R14del-Phospholamban Cardiomyopathy"

_ijms, 2023, doi:10.3390/ijms24032675_

Round 1

Reviewer 1 Report

In the manuscript 'Myofilament Alterations Associated with Human R14del-Phospholamban Cardiomyopathy' submitted by Kumar et al. to the International Journal of Molecular Sciences, the authors investigated the molecular defects caused by a small deletion in the PLN gene leading to arrhythmogenic cardiomyopathy.

Although the manuscript is interesting it needs some small corrections:

1.) Please use the official nomenclature for the mutations in the complete manuscript: See Mutalyzer 3

2.) Please write all human gene names in Italics.

3.) Please give a short overview about other genes involved in arrhythmogenic cardiomyopathy and add a relevant review articel for example: Insights Into Genetics and Pathophysiology of Arrhythmogenic Cardiomyopathy Brenda GerullAndreas Brodehl; PMID: 34478111

  • PMCID: PMC8616880
  •  

DOI: 10.1007/s11897-021-00532-z

4.) Please add the OMIM identifier number for ARVC/ ACM in line 41.

5.) Some references are highlighted in red color. Why?

6.) Please write in the complete manusript Ca2+ rather than Ca.

 7.) Please change SEM against SD since the reader is interested in variation of your data. Larger error bars should not be a problem :-).

8.) Please increase the size of the Figure 2 and Figure 3 by other formation. 

9.) Please increase the quality of the echocardiography images in Figure 6. They are to small.

10.) Scale bars are missing in Figure 7,8,9.

Good luck with the revision!

Author Response

Reviewer 1

 Please use the official nomenclature for the mutations in the complete manuscript: See Mutalyzer 3

RESPONSE: We have now introduced the Mutalyzer 3 nomenclature as NM_002667.5:c.40_42del (Mutalyzer 3) on lines 53 and 54.

Please write all human gene names in Italics.

RESPONSE: We have now followed the italics format throughout the manuscript.

Please give a short overview of other genes involved in arrhythmogenic cardiomyopathy and add a relevant review article for example: Insights Into Genetics and Pathophysiology of Arrhythmogenic Cardiomyopathy Brenda Gerull, Andreas Brodehl; PMID: 34478111, PMCID: PMC8616880, DOI: 10.1007/s11897-021-00532-z

RESPONSE: We have included an overview of other genes involved in arrhythmogenic cardiomyopathy in the first paragraph of the Introduction of the revised manuscript: Lines 46-52. As suggested, we have also added the relevant review article (ref # 3).

Please add the OMIM identifier number for ARVC/ ACM in line 41.

RESPONSE: We have now added the OMIM number 609040 in line 43.

Some references are highlighted in red color. Why?

RESPONSE: They were typos. We are sorry about that. In the revised version, the references are not highlighted in red color.

Please write in the complete manuscript Ca2+ rather than Ca.

RESPONSE: We have now adopted the Ca2+ format throughout the manuscript.

Please change SEM against SD since the reader is interested in a variation of your data. Larger error bars should not be a problem :-).

RESPONSE: We thank the reviewer for the suggestions. We have now revised the figures with SD and presented them in the revised manuscript.

Please increase the size of Figure 2 and Figure 3 by other formations. 

RESPONSE: We have now increased the font size for figures 2 and 3.

Please increase the quality of the echocardiography images in Figure 6. They are to small.

RESPONSE: We have now increased the quality of the echocardiography images in figure 6.

Scale bars are missing in Figure 7,8,9.

RESPONSE: We have now added scale bars in figures 7-9.

Reviewer 2 Report

The study provided by Kumar et al. investigates cardiac effects by myofilament alterations with a specific R14del-PLN mutation in vivo and in vivo. The study is of high interest, the manuscript is well written and the methodology is sound and extensive.

Some minor points need to be clarified:

1. Showing representative Ca transients of the isolated myocytes would greatly improve the manuscript.

2. Figure 1 D+H, although bar graphs in H show significant differences in pCA between WT and mutant, the related curves in D) do, in my opinion, show now differences. Can you please comment on this?

3. The shown changes in Ca-homeostasis are quite drastic. Although this paper focusses more on structural changes, one is wondering wether the isolated cardiomyocytes or the mice develop either spontanous activation or are more prone to induced ventricular tachycardias (the enhanced interstitital fibrosis might point to that as well). Can you provide data about spontanous activity or VTs, or comment on this? That would fit into the echo data showing no differences in baseline heart function.

4. Could you please provide some more information between the link of altered Ca-homeostasis and the observed structural changes? Do the observed effects possibly lead to MMP or fibroblast activation, which could explain the increase of ventricular fibrosis.

5. Could you please provide more information regarding the echo analysis? Describing EF, FS etc. calculation would improve the method section.

Author Response

Showing representative Ca transients of the isolated myocytes would greatly improve the manuscript.

RESPONSE: We have added the Ca2+ transients in the revised figure 5.

Figure 1 D+H, although bar graphs in H show significant differences in pCA between WT and mutant, the related curves in D) do, in my opinion, show now differences. Can you please comment on this?

RESPONSE: We thank the reviewer for this question. The bar graph axis range is made small to show the difference. The curve has a long range on both the x & Y axis. Therefore, it looks compressed.  In literature, the nH is well established (PMID: 26453301, PMID: 24464755 and PMID: 25463273) by the first and last authors.

The shown changes in Ca-homeostasis are quite drastic. Although this paper focusses more on structural changes, one is wondering wether the isolated cardiomyocytes or the mice develop either spontanous activation or are more prone to induced ventricular tachycardias (the enhanced interstitital fibrosis might point to that as well). Can you provide data about spontanous activity or VTs, or comment on this? That would fit into the echo data showing no differences in baseline heart function.

RESPONSE: The isolated cardiomyocytes or intact mice do not show spontaneous after-contractions or ventricular arrhythmias. These are only observed under stress conditions at the cellular and whole animal levels. This point is clarified in the Discussion of the revised manuscript lines 364-370.

Could you please provide some more information between the link of altered Ca-homeostasis and the observed structural changes? Do the observed effects possibly lead to MMP or fibroblast activation, which could explain the increase of ventricular fibrosis.

RESPONSE: We thank the reviewer for raising this point. We have addressed the link between cardiomyocyte calcium homeostasis and fibroblast activation in the revised manuscript under Discussion (lines 366-373) and provide a relevant review reference (ref # 28).

Could you please provide more information regarding the echo analysis? Describing EF, FS etc. calculation would improve the method section.

RESPONSE: More details were included in the Methods Section under Echocardiography, as suggested by the reviewer, with two references (Ref # 37 and 38).